# Arterial Stiffness following Endurance and Resistance Exercise Sessions in Older Patients with Coronary Artery Disease

**DOI:** 10.3390/ijerph192214697

**Published:** 2022-11-09

**Authors:** Vanessa Santos, Luís Miguel Massuça, Vitor Angarten, Xavier Melo, Rita Pinto, Bo Fernhall, Helena Santa-Clara

**Affiliations:** 1CIPER, Exercise and Health Laboratory, Faculdade de Motricidade Humana, Universidade de Lisboa, Cruz Quebrada, 1649-004 Lisbon, Portugal; 2KinesioLab, Research Unit in Human Movement Analysis, Instituto Piaget, 2805-059 Almada, Portugal; 3ICPOL Research Center, Higher Institute of Police Sciences and Internal Security, 1349-040 Lisbon, Portugal; 4CIDEFES—Research Center in Sport, Physical Education, Exercise and Health, Lusófona University, 1749-024 Lisbon, Portugal; 5Egas Moniz Interdisciplinary Research Center (CiiEM), Egas Moniz School of Health, 2829-511 Almada, Portugal; 6Structural and Coronary Heart Disease Unit, Centro Cardiovascular da Universidade de Lisboa (CCUL@RISE), Faculty of Medicine, University of Lisbon, 1649-004 Lisbon, Portugal; 7College of Nursing and Health Sciences, University of Massachusetts Boston, 100 Morrissey Boulevard, Boston, MA 02125-3393, USA

**Keywords:** acute exercise adaptations, arterial stiffness, cardiovascular disease, pulse wave velocity

## Abstract

Arterial stiffness (AS) is associated with coronary artery disease (CAD). Acute endurance training decreases AS, whereas acute resistance training increases it. However, these results are from studies in apparently healthy adults, and there is no information on the effects of such afterload AS in elderly patients with CAD. We aimed to investigate the effect of acute endurance or resistance training on the time course of changes in the indices of AS in elderly patients with CAD in order to understand how stiffness responds after training. We tested 18 trained men with CAD. AS was measured using central and peripheral pulse wave velocity (PWV) after 15 min of rest and after 5, 15, and 30 min of endurance and resistance training sessions. The endurance session consisted of high-intensity interval walking at 85–90% of maximum heart rate, and the resistance session consisted of 70% of the maximum of one repetition. An interaction effect was found for central and peripheral PWV (*p* ≤ 0.001; carotid, η^2^ = 0.72; aortic, η^2^ = 0.90; femoral, η^2^ = 0.74), which was due to an increase in PWV after resistance and a decrease in central and peripheral PWV after endurance. This study demonstrates that training mode influences the time course of AS responses to acute exercise in these patients. Acute endurance training decreased AS, whereas resistance training significantly increased it.

## 1. Introduction

Arterial stiffness (AS) is closely related to cardiovascular disease (CVD) and is an important marker of mortality [1,2]. Large artery stiffness is a central paradigm of aging, but the extent of increase may depend on various environmental or genetic factors [3]. The underlying vascular pathology associated with coronary artery disease (CAD) results from common biochemical associations with myocardial stiffness, such as abnormal collagen turnover, cytokines, metalloproteinase activity, and increased vascular tone [4].

Pulse wave velocity (PWV) is a non-invasive method used to assess AS [5,6,7]. Aortic PWV has clinical relevance to cardiovascular prognosis and is a predictor of functional capacity [5,6,7,8,9,10,11,12,13], future cardiovascular events [2], and mortality [5,6,7]. Independent of traditional risk factors, an increase of 1 m/s in aortic PWV increases total cardiovascular events, cardiovascular mortality, and all-cause mortality by 14–15% [13]. In addition to its clinical value, aortic PWV also responds to pharmacological [14,15] and physical therapies [16,17]. Several factors influence AS, including blood pressure (BP), sympathetic activity, and vasomotor tone. These factors are also influenced by both physical activity and pharmacologic therapy. For example, changes in BP, sympathetic drive, and vasomotor tone can be influenced by the mode of exercise [18] and intensity [6], the site of arterial stiffening [19], the frequency of measurements [18,20], and the studied population itself [21,22]. As an example, in apparently healthy subjects, a single bout of endurance training has been shown to increase AS [17,23,24,25,26], whereas acute endurance training has been shown to decrease AS [16,27]. Therefore, heterogeneity is evident within all comparisons, with several methodological concerns noted, including the low applicability of exercise protocols and the lack of a control intervention.

Although CAD remains the leading cause of death in individuals over 30 years of age [3,28] and physical training can attenuate disease progression and improve survival in the secondary prevention of CAD [6,7], only two studies have examined the acute effects of exercise on AS in patients with CAD [29,30]. In these studies, the authors showed that brief treadmill exercise can effectively improve AS in patients with CAD, but only brachial ankle PWV (a measure of combined aortic and peripheral stiffness) was used to measure stiffness. Unfortunately, central BP and aortic stiffness were not measured, and PWV after exercise was measured only once after a single treadmill test. A recent systematic review and meta-analysis of 45 studies suggests that the type of exercise is an important determinant of AS in apparently healthy participants after brief exercise training, with marked differences in responses after acute aerobic endurance training and resistance training. However, there are no previous studies that have directly compared exercise-induced changes in AS after different types of exercise, with discrete responses after each type of exercise in patients with CAD. Although laboratory approaches are widely used in exercise research, an obvious drawback is their low ecological validity [31,32]. A better understanding of arterial responses to different types of exercise, as typically encountered in exercise-based cardiac rehabilitation units, could help select appropriate exercise protocols to improve cardiovascular function and help identify periods of significant concern that could be caused by transient changes in PWV immediately after exercise.

Therefore, the aim of this study was to compare the time course of changes in local and regional indices of AS after endurance exercise training (EES) and resistance training (RES) in the cardiac rehabilitation of patients with CAD. Our hypothesis was that AS increases after resistance training and decreases after endurance training.

## 2. Materials and Methods

### 2.1. Participants

Thirty patients with CAD, who had been exercising for at least 6 months in an exercise-based cardiac rehabilitation program, were invited to participate in this study. The inclusion criteria were as follows: the patients were receiving optimal medical therapy for CAD with a stable condition for more than 1 month (no hospitalization and no change in medication). The exclusion criteria included more than one of the following risk factors: current smoking (<6 months), diabetes, impairment of musculoskeletal mobility, or taking vasodilators. Of these 30 patients, 6 did not complete both sessions, 1 did not meet the inclusion criteria, and 5 refused to participate. A total of 18 older adults aged 71.8 ± 10.2 years (ranging from 65.0 to 82.0 years) with a stable CAD diagnosis had complete data for both training sessions.

All subjects provided written informed consent prior to participation. The Faculty of Human Kinetics Ethics Committee approved this study, and all procedures and the treatment of subjects were in accordance with the Declaration of Helsinki.

### 2.2. Study Design

This was a randomized, cross-over, repeated-measures study. Participants attended 2 separate interventions consisting of EES and RES exercise sessions at least 48 h apart. Body composition measurements were taken a week before. Participants had been familiar with this type of exercise session for at least 6 months. Participants were assessed at rest (15 min supine, in a quiet, dimly lit, temperature-controlled laboratory) and 5, 15, and 30 min after the exercise session. These time points were selected based on previous results [33,34]. Heart rate was continuously monitored throughout the exercise session (Polar monitor), and perceived exertion (RPE) was assessed [35].

All patients performed a maximal cardiopulmonary exercise test according to the Bruce protocol.

All sessions were performed in the morning, with each participant completing the sessions at the same time of day to minimize potential diurnal variation. Participants were instructed not to consume any food or drink other than water after midnight before the sessions and to abstain from alcohol, caffeine, and exercise for at least 24 h before each session. The same researchers collected all the data, and the patients were blinded to the order of the exercise sessions.

### 2.3. Exercise Sessions

The EES and RES exercise sessions were designed to represent typical training sessions in accordance with the current guidelines for people with CVD [36,37]. The endurance training sessions consisted of a 5 min warm-up at 60% of maximum heart rate followed by 25 min of supervised treadmill walking, using a high-intensity interval training method. The main part of the training sessions consisted of 10 interval training periods (2 min at 85–90% of maximum heart rate) and 9 rest periods (1 min passive rest) between the interval training periods. This training method was chosen because there is increasing evidence that high-intensity interval training is more effective than continuous moderate-intensity training in improving cardiorespiratory fitness in cardiac patients [38,39]. Resistance training sessions consisted of a warm-up program of 2 upper-limb exercises and 2 lower-limb exercises repeated 12 times at 30–40% of a maximum repetition, and then 3 sets of 8 repetitions of strength exercises on 6 weight machines (3 for the upper limbs and 3 for the lower limbs) performed at 70% of a maximum repetition. Participants were given a 30 s rest between exercises and a 1 min rest between sets. The maximum repetitions were determined beforehand. Heart rate was continuously monitored, as well as RPE at the end of each exercise, and exercise physiologists ensured that the Valsalva maneuver was avoided and that the patients exercised at the prescribed intensities.

### 2.4. Central and Peripheral Arterial Stiffness Indices

Blood pressure (BP) was measured twice on the right arm with an automatic cuff (OMRON 907). If the difference in systolic BP between the two measurements was greater than 10 mmHg, another BP measurement was performed. Mean arterial pressure [MAP = diastolic BP + 1/3 (systolic BP − diastolic BP)] and pulse pressure (PP = systolic BP − diastolic BP) were calculated for fitting purposes.

The stiffness of the right carotid artery was measured in the supine position after at least 15 min of rest before and within 5, 15 and 30 min after each session. We used an ultrasound machine equipped with a 13 MHz linear probe (MyLab One, Esaote, Genova, Italy) with Quality Arterial Stiffness technology based on a radiofrequency signal in a segment of the right carotid artery ~1 cm anterior to the bifurcation. The right carotid artery pressure curve was calibrated to the right brachial artery diastolic pressure by iteratively changing the wall stiffness coefficient and MAP. This allows for the calculation of carotid PWV (m/s) using the following equation: $\mathrm{PWV}=\frac{1}{\sqrt{\rho \cdot DC}}=\sqrt{\frac{D^{2}\cdot \Delta PP}{\rho \cdot \left(2\cdot D\cdot \Delta D+\Delta D^{2}\right)}}$, where *D*: diastolic diameter; Δ*D*: change in diameter in systole; *DC*: coefficient of distensibility; Δ*PP*: local pulse pressure; and *ρ*: blood density [40]. PWV was also measured using applanation tonometry immediately after ultrasonography. A single operator located the carotid, femoral, and distal posterior tibial arteries on the right side of the body and marked the point for acquisition of the corresponding pressure curves with two specific pressure-sensitive transducers. The distance between the carotid, femoral, radial, and distal posterior tibial arteries was measured directly and entered into Complior Analysis software using a correction factor of 0.8 (ALAM Medical, Paris, France). The right brachial BP was measured and entered into the software, and then signal acquisition was started. Duplicate values obtained from the femoral and posterior tibial carotid arteries were used as indices of central/aortic and peripheral AS, respectively. Whenever a continuous decrease before the sharp systolic upstroke was not clearly evident or the tolerance exceeded 0.5 m/s, a third second measurement was taken.

### 2.5. Statistical Analysis

The power and sample size calculations (G-Power, Version 3.1.3) were based on a predicted central PWV difference of 0.9 m/s with an SD of 0.2 m/s [41], α = 0.05, 1 − β = 0.80, and an expected dropout rate of 20%. According to these calculations, 14 patients were required in each session. Comparisons between interventions (EES vs. RES) over time (rest vs. post-intervention time points) for AS indices were examined using a 2-way repeated-measures analysis of variance in two directions (intervention × time) (ANOVA). The Bonferroni test was used for a post hoc comparison of the means between each pair of groups. Because AS depends on arterial pressure [42], the percentage of change in MAP from rest to 30 min after exercise was included as a covariate.

## 3. Results

The characteristics of the patients with CAD, who participated in the study, are shown in Table 1. All participants reached 85–90% of peak heart rate during the EES sessions and scored 7–9 on the 10-point RPE scale [43], corresponding to severe effort (Table 2).

An intervention by the time interaction effect (Figure 1) was associated with a large size effect observed for the carotid (η^2^ = 0.72, *p* < 0.001), aortic (η^2^ = 0.90, *p* < 0.001), and femoral PWV (η^2^ = 0.74, *p* < 0.001), as PWV decreased following EES at 5 min (carotid: d = 0.49, *p* < 0.01; aortic: d = 0.53, *p* < 0.01), 15 min (carotid: d = 0.89, *p* < 0.01; aortic: d = 0.87, *p* < 0.01; femoral: d = 0.46, *p* < 0.01), and 30 min (carotid: d = 1.21, *p* < 0.01; aortic: d = 0.91, *p* < 0.01; femoral: d = 0.53, *p* < 0.01) into recovery, whereas PWV increased following RES at 5 min (carotid: d = −0.98, *p* < 0.01; aortic: d = −0.58, *p* < 0.01; femoral: d = −0.51, *p* < 0.01), 15 min (carotid: d = −1.40, *p* < 0.01; aorta: d = −1.31, *p* < 0.01; femoral: d = −0.71, *p* < 0.01), and 30 min (carotid: d = −0.76, *p* < 0.01; aorta: d = −1.41, *p* < 0.01; femoral: d = −0.19, *p* = 0.05) into recovery (Figure 1A–C). We also found an intervention by the time interaction effect on heart rate recovery (Table 2) (η^2^ = 0.63, *p* < 0.001), as the percentage decrease in HR recovery was higher at minutes 5 (d = −21.2, *p* < 0.01) and 15 (d = −11.6, *p* < 0.01) after RES than after EES. No significant interaction effects were found between intervention and time or the main effects of time on the indices of central or brachial BP (Table 3).

Controlling PWV for changes in MAP did not change the results.

Changes in carotid PWV from rest to minute 30 were positively associated with carotid PWV at rest and with femoral PWV (*p* < 0.05) in RES. Changes in aortic PWV from rest to minute 30 were positively associated with aortic PWV at rest, and changes in aortic PWV from rest to minute 5 were inversely associated with central MAP (*p* < 0.05) in EES. Changes in femoral PWV were (i) positively associated with carotid PWV at rest from rest to minute 30 in EES, (ii) positively associated with carotid PWV at rest from rest to minute 5 in RES, and (iii) associated with femoral PWV at rest from rest to minute 15 (*p* < 0.05) (Table 4).

## 4. Discussion

To the best of our knowledge, this is the first study to directly compare changes in local and regional indices of arterial stiffness (AS) after exercise in response to EES and RES in patients with CAD. We demonstrated a significant decrease in vascular stiffness throughout the arterial tree at 5, 15, and 30 min after EES, while an increase in vascular stiffness was observed after RES. It remains unclear whether the increase in AS after RES is of clinical significance in patients with CAD. The absence of cardiovascular events could be due to the small size of the studied population, the short follow-up period, increased cardiac perfusion during RES, or diastolic BP during RES [18].

### 4.1. Arterial Stiffness Indices Immediately after Exercise (5 min)

The acute effects of exercise on AS in apparently healthy adults during the first few minutes of recovery remain controversial. Some studies have shown no changes in AS indices in the first 5 min after acute endurance training [45,46,47,48,49,50,51,52,53] or resistance training [17,54,55], whereas others have reported significant changes in both central and peripheral AS indices [47,53]. To date, few studies have been conducted in patients with clinical conditions, especially CVD. Angarten et al., 2021, demonstrated an increase in aortic stiffness immediately after a cardiopulmonary exercise test (10 min of recovery) in participants with and without CAD, and they stated that an increase in PWV of ~0.8 m/s in patients with and without CAD should not be ignored because it may increase the risk of sudden cardiac events [56]. Alonso-Domínguez et al., 2019, reported that AS did not change after endurance exercise in patients with type 2 diabetes [57], and, more recently, Trachsel et al., 2019, showed that the immediate decrease in femoral PWV during a maximal cardiopulmonary exercise test with an individualized ramp protocol observed in healthy young participants was absent in patients with CAD and in apparently healthy peers [29]. This could be explained by age differences, as the patients with CAD in the study conducted by Trachsel et al., 2019, had a mean age of 55 years, and the population in our study was older (age of 71.8 ± 10.2 years), possibly with more severe disease [29]. The intensity and duration of effort were also different, as the endurance component in their study was maximal exercise testing [29] and that in ours was high-intensity interval training, which may explain the differences. Our results were also adjusted for BP. Regarding the acute effect of resistance training, in contrast to our study, Heffernan et al., 2007, and Thiebaud et al., 2016, showed no significant differences 5 min post-exercise in aortic PWV in healthy young participants [55,58]. Mak and Lai, 2015, also showed no significant differences in aortic PWV immediately after exercise in healthy young men [23].

### 4.2. Arterial Stiffness Indices after Exercise (15–30 min)

There is a general trend in the literature for AS indices (aortic, brachial, and femoral stiffness) to decrease to resting values or below in apparently healthy adults after endurance training [16,17,41,48,49,50,51,53,58,59]. However, for studies using the acute resistance exercise model, the evidence is somewhat mixed. Yoon et al., 2010, showed no differences in aortic PWV at 20 min [24], whereas increases in aortic PWV at 10 min [25] and 20 min [24] have also been reported. Regarding patients with clinical conditions, Lefferts et al., 2018, showed that aortic PWV remained elevated in the first 10 min after endurance training in patients with hypertension [60], and a small number of studies were performed with resistance training. Fahs et al., 2009, showed a significant increase in aortic PWV 15 min after resistance training in healthy young men [26], and Heffernan et al., 2007, also showed a significant increase in aortic PWV 20 min after resistance training in healthy young men [17].

In patients with CAD, Sung et al., 2009, observed a decrease in brachial ankle PWV 10 min after exercise [30], and a more recent study conducted by Trachsel et al., 2019, showed that femoral PWV, measured between the right femoral artery and the right posterior tibial artery, increased between 5 and 14 min after exercise [29].

Although the exact mechanisms that regulate the modal differences in aortic PWV and femoral PWV have not yet been identified, several mechanisms have been considered [24,27], with differential adaptations to BP during exercise, particularly during RES, likely playing an important role in the immediate changes in aortic PWV [17]. However, we did not detect significant changes in central and peripheral BP throughout the post-exercise period. However, BP was probably different during exercise, and this could have an important influence. Heffernan et al., 2007, investigated the independent role of the Valsalva maneuver as a stimulus contributing to arterial wall stiffening, which is often attributed to RES alone [61]. The authors showed that, with a similar increase in BP during exercise, only the groups performing the Valsalva maneuver had increased vessel wall thickness [61]. Changes in endothelial cell signaling and vascular smooth muscle can also influence vascular stiffness [61]. Changes in pressure can apparently lead to changes in stiffness, provided that the volume remains stable, and, as shown by Heffernan et al., 2007, not only can the Valsalva maneuver alter the relationship between pressure and volume, but it can also affect vascular properties [61].

The complex relationship between acute EES and RES and changes in AS in patients with CAD has been demonstrated, namely, that segments of the arterial tree are affected differently and respond/recover in a time-dependent manner after the cessation of exercise. However, our results suggest a harmonic response throughout the arterial tree that cannot be predicted by AS indices at rest. We can only speculate that the modal oscillations of PWV depend on different sympathetic and vagal modulatory influences (Lefferts et al., 2018 [60]). However, aortic stiffness is not directly regulated by the autonomic nervous system, although it is influenced by changes in heart rate and MAP [62]. Recently, Holwerda et al., 2019, demonstrated that central AS is influenced by muscle sympathetic nerve activity [63] and likely contributes to the age-related increase in stiffness throughout the arterial tree. Further studies are needed to understand this influence in detail. However, the hemodynamic response in the present study cannot explain the modal differences in PWV, the decreases after EES, and the increases after RES. Understanding these responses may help in the selection of appropriate exercises to improve cardiovascular function, particularly for these at-risk populations. The relationship between the type of exercise and the measures of AS in elderly patients with CAD is complex, and although the current results exceed those of previous studies, a better understanding of the physiological responses to acute exercise is warranted.

This study is not without limitations. The sample studied consisted only of men, as the rehabilitation programs in which recruitment took place are predominantly attended by men. Sex differences have been reported both in resting aortic PWV and after acute endurance exercise [64]. This limits the generalization of the results to the entire population with CAD. The order of experimental interventions was not communicated to participants in a blinded fashion because of limited options. Another important limitation was that monitoring post-intervention AS at designated time points for the first 30 min may have failed to identify subtle changes or those outside this initial period (e.g., 1–72 h).

## 5. Conclusions

The present study demonstrated that the mode of exercise influences the time course of the response of local and regional AS to acute exercise in elderly patients with CAD. Acute endurance training decreased central and peripheral AS, while acute resistance training significantly increased central and peripheral AS in patients with CAD. This study gives us a better understanding of acute responses to exercise, which can help researchers and clinicians to select the appropriate mode of exercise. This is critical for patients with CAD aiming to prevent the risk and/or occurrence of a critical cardiovascular event.

Additionally, we demonstrated that post-exercise measurements of AS can be used to assess the intrinsic stiffness of the central and peripheral arteries in patients with CAD.

## Figures and Tables

**Figure 1 ijerph-19-14697-f001:**
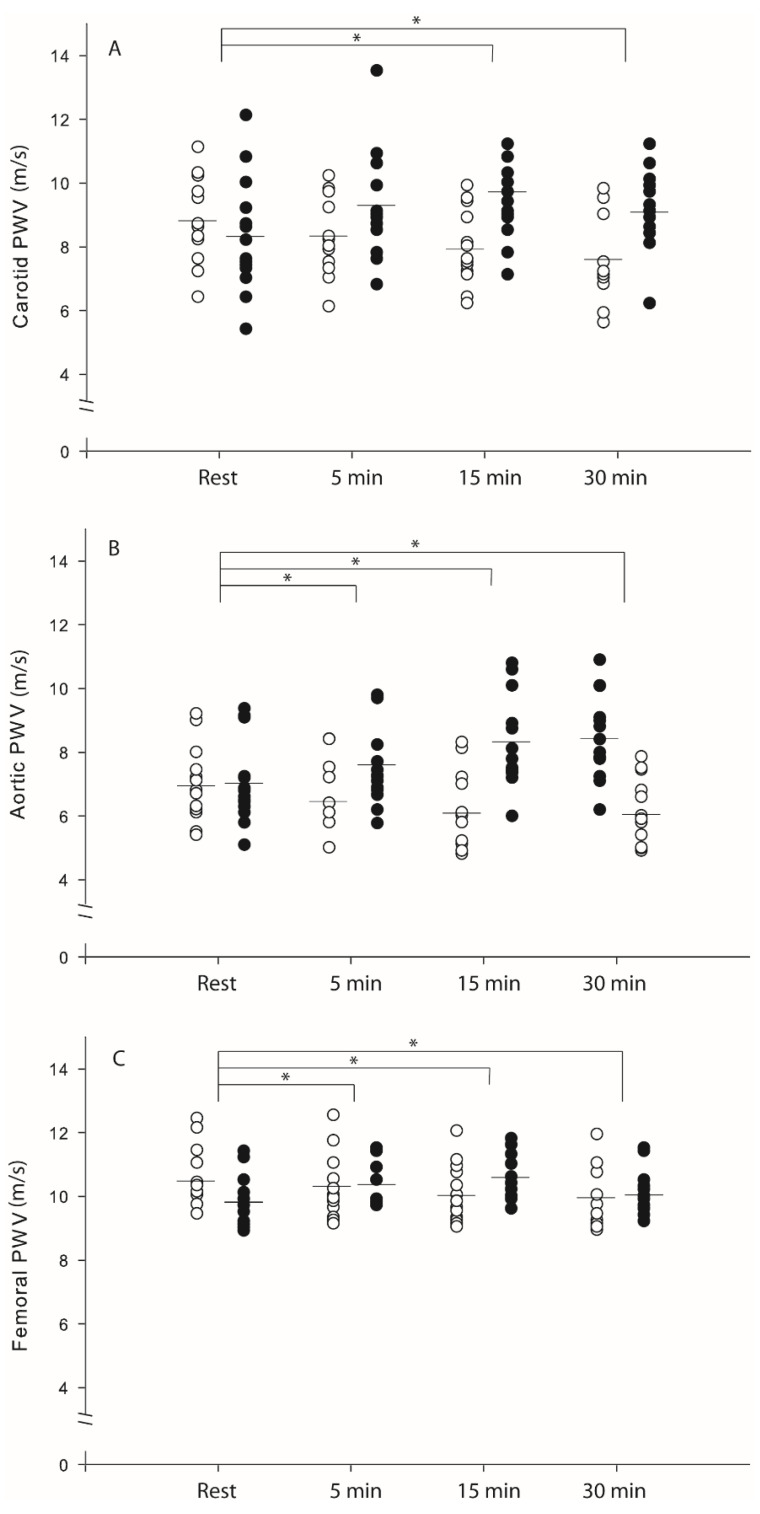
Carotid (**A**), aortic (**B**), and femoral (**C**) PWV at rest and throughout the post-exercise period. Data are presented as (◯) EES and (⬤) RES. * Significant difference from baseline within group: *p* < 0.05.

**Table 1 ijerph-19-14697-t001:** Characteristics of the patients.

Variables	*n* = 18 Patients
Age (years)	71.8 ± 10.2
Sex	Male
Weight (kg)	74.0 ± 9.9
Height (m^2^)	1.68 ± 0.1
BMI (kg/m^2^)	26.2 ± 2.8
Brachial SBP at rest (mmHg)	119.9 ± 12.4
Brachial DBP at rest (mmHg)	65.4 ± 8.4
Hypertension (%)	79
Hyperlipidemia (%)	86
Overweight/obesity (%)	28
>1 year ex-depression (%)	33
CAD summary	
>1 year CABG (%)	0
>1 year diagnosis (%)	10
PCI (%)	86
STEMI (%)	86
Previous MI (%)	86
Medication	
Beta-blocker (%)	100
ACEI/ARB (%)	78
Statin (%)	78
Antiplatelet (%)	100
Diuretics (%)	64
Calcium channel blockers (%)	43

ACEI/ARB—angiotensin-converting enzyme inhibitor/angiotensin receptor blocker; BMI—body mass index; CABG—coronary artery bypass graft; DBP—diastolic blood pressure; MI—myocardial infarction; PCI—percutaneous coronary intervention; SBP—systolic blood pressure; STEMI—ST-elevation myocardial infarction.

**Table 2 ijerph-19-14697-t002:** Heart rate session variables.

	EES	RES
Heart rate rest (bpm)	62 ± 5	62 ± 5
Peak heart rate (bpm)	113 ± 13 *	87 ± 8 *
TRIMP	104.9 ± 9.8	43.3 ± 10.6
RPE	8 ± 1	8 ± 1
Heart rate recovery 5 min (bpm)	104 ± 9 *	83 ± 8 *
Heart rate recovery 15 min (bpm)	89 ± 4 *	77 ± 7 *
Heart rate recovery 30 min (bpm)	63 ± 4 *	64 ± 4 *

*, *p* < 0.01 (significantly different from rest); bpm—beats per minute; EES—endurance exercise session; RES—resistance exercise session; RPE—rated perceived exertion; TRIMP—training impulses, according to the method of Edwards [44].

**Table 3 ijerph-19-14697-t003:** Arterial stiffness indices at rest and following endurance and resistance exercise sessions.

Variables	Sessions	Rest	5 min after	15 min after	30 min after	Main Effect of the Session	Main Effect of Time
Heart rate (% of difference)	EES	-	−17.7 ± 4.0	−29.7 ± 7.5	−49.2 ± 8.5	RES > EES *	rest < 5, 15 5 > 15, 30 15 > 30 *
	RES	-	−33.6 ± 10.6	−38.5 ± 9.4	−48.7 ± 7.6		rest < 5, 15 5 > 15, 30 15 > 30 *
Brachial SBP (mmHg)	EES	120.0 ± 11.3	117.9 ± 19.5	107.9 ± 11.3	109.0 ± 11.7	NS	NS
	RES	120.9 ± 13.8	123.1 ± 17.4	119.8 ± 10.2	120.0 ± 12.0		NS
Brachial DBP (mmHg)	EES	65.0 ± 8.0	68.5 ± 12.1	64.4 ± 7.4	62.1 ± 7.4	NS	NS
	RES	65.7 ± 9.0	63.6 ± 9.9	63.1 ± 10.2	63.9 ± 9.8		NS
Brachial MAP (mmHg)	EES	82.9 ± 7.6	84.7 ± 13.6	78.7 ± 8.0	77.8 ± 7.9	NS	NS
	RES	83.9 ± 8.8	83.1 ± 9.7	81.7 ± 8.5	82.4 ± 8.8		NS
Brachial PP (mmHg)	EES	54.1 ± 11.5	49.4 ± 12.8	43.8 ± 8.8	46.5 ± 9.9	NS	NS
	RES	55.1 ± 13.9	59.5 ± 18.1	56.6 ± 11.8	56.2 ± 12.2		NS
Central SBP (mmHg)	EES	109.5 ± 10.5	108.9 ± 15.4	103.1 ± 12.4	103.6 ± 11.8	NS	NS
	RES	113.4 ± 13.4	111.8 ± 16.0	109.9 ± 9.5	110.2 ± 8.0		NS
Central DBP (mmHg)	EES	65.4 ± 6.6	65.7 ± 13.1	64.6 ± 9.5	62.6 ± 9.7	NS	NS
	RES	65.7 ± 9.0	62.0 ± 10.6	62.9 ± 10.6	63.4 ± 9.9		NS
Central MAP (mmHg)	EES	82.6 ± 6.7	82.9 ± 13.1	79.4 ± 9.2	78.2 ± 9.0	NS	NS
	RES	83.8 ± 8.9	82.7 ± 10.5	81.6 ± 8.7	81.6 ± 8.2		NS
Central PP (mmHg)	EES	44.1 ± 8.3	43.2 ± 10.7	38.9 ± 11.7	41.3 ± 10.6	NS	NS
	RES	47.6 ± 14.2	49.9 ± 16.9	47.0 ± 13.4	46.9 ± 10.7		NS

*, *p* < 0.05 significantly different from rest value; 15—15 min post-session; 30—30 min post-session; 5—5 min post-session; DBP—diastolic blood pressure; EES—endurance session; MAP—mean arterial pressure; NS—non-significant; PP—pulse pressure; RES—resistance session; SBP—systolic blood pressure.

**Table 4 ijerph-19-14697-t004:** Correlation coefficients for central and peripheral PWV post-exercise.

	Pearson Correlation Coefficient (*p*-Value)					
	EES			RES		
	5 min	15 min	30 min	5 min	15 min	30 min
Δ PWV Carotid						
Brachial SBP	0.32 (0.26)	0.28 (0.34)	0.14 (0.64)	−0.15 (0.62)	−0.23 (0.43)	0.04 (0.90)
Brachial DBP	0.18 (0.54)	0.05 (0.86)	0.11 (0.70)	0.05 (0.86)	−0.03 (0.92)	−0.26 (0.06)
Brachial PP	0.19 (0.51)	0.24 (0.41)	0.06 (0.85)	−0.18 (0.54)	−0.21 (0.47)	0.43 (0.13)
Brachial MAP	0.30 (0.30)	0.19 (0.52)	0.16 (0.60)	0.03 (0.91)	−0.14 (0.64)	−0.41 (0.14)
Carotid PWV	0.39 (0.17)	0.48 (0.08)	0.11 (0.70)	0.37 (0.19)	0.19 (0.52)	0.72 (<0.001) *
Aortic PWV	0.05 (0.86)	0.19 (0.52)	−0.03 (0.92)	0.34 (0.23)	0.17 (0.56)	0.52 (0.06)
Femoral PWV	0.15 (0.60)	0.20 (0.50)	−0.02 (0.96)	0.11 (0.72)	−0.12 (0.70)	0.68 (0.01) *
Δ PWV Aortic						
Brachial SBP	−0.43 (0.12)	−0.51 (0.06)	−0.25 (0.39)	−0.23 (0.42)	−0.07 (0.81)	0.014 (0.63)
Brachial DBP	−0.37 (0.20)	−0.41 (0.15)	−0.06 (0.83)	0.06 (0.84)	0.10 (0.74)	−0.00 (0.99)
Brachial PP	−0.17 (0.56)	−0.22 (0.45)	−0.20 (0.49)	−0.27 (0.35)	−0.13 (0.65)	0.14 (0.63)
Brachial MAP	−0.47 (0.09)	−0.54 (0.05) *	−0.17 (0.57)	−0.09 (0.77)	0.03 (0.92)	0.07 (0.82)
Carotid PWV	0.03 (0.93)	−0.15 (0.62)	0.05 (0.88)	−0.02 (0.96)	−0.04 (0.89)	0.09 (0.76)
Aortic PWV	0.39 (0.17)	0.13 (0.66)	0.54 (0.05) *	0.07 (0.81)	−0.04 (0.90)	0.22 (0.45)
Femoral PWV	−0.02 (0.94)	−0.23 (0.44)	0.15 (0.62)	−0.05 (0.85)	−0.11 (0.72)	−0.30 (0.92)
Δ PWV Femoral						
Brachial SBP	0.21 (0.48)	0.23 (0.42)	−0.09 (0.76)	0.17 (0.56)	0.23 (0.43)	−0.09 (0.75)
Brachial DBP	0.28 (0.33)	0.35 (0.22)	0.41 (0.14)	−0.36 (0.21)	−0.49 (0.07)	−0.45 (0.10)
Brachial PP	0.01 (0.98)	−0.01 (0.97)	−0.38 (0.18)	0.40 (0.16)	0.25 (0.14)	0.20 (0.49)
Brachial MAP	0.32 (0.27)	0.37 (0.20)	0.26 (0.36)	−0.18 (0.55)	−0.24 (0.41)	−0.38 (0.19)
Carotid PWV	0.28 (0.33)	0.30 (0.30)	0.68 (0.01) *	0.57 (0.03) *	0.52 (0.06)	0.41 (0.15)
Aortic PWV	−0.15 (0.61)	−0.09 (0.76)	0.09 (0.77)	0.50 (0.07)	0.41 (0.15)	0.35 (0.22)
Femoral PWV	−0.02 (0.95)	0.36 (0.20)	0.32 (0.26)	0.54 (0.04) *	0.62 (0.02) *	0.37 (0.19)

* *p* < 0.05 significantly different.

## Data Availability

All data relevant to the study are included in the article.

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
