# Peer review of "Arterial Stiffness following Endurance and Resistance Exercise Sessions in Older Patients with Coronary Artery Disease"

_ijerph, 2022, doi:10.3390/ijerph192214697_

Round 1
Reviewer 1 Report
Dear Authors, the paper is well written.
I found it very interesting. I appreciated the idea of the paper, but there are some points that need to be clarified:
- How was ejection fraction of patients enrolled? and NYHC classification?
- Did you excluded patients with atherosclerosis of carotid, femoral or aortic vassel? If yes specified in the exclusion section
- Could you report the mid-intimal thickness of the vessel of the carotid, brachial and femoral artery?
-In the drugs section of patients could you report how many patients take vasodilators (Peripheral calcium antagonists, nitrates and so on)
Author Response
Arterial stiffness following endurance and resistance exercise sessions in older patients with coronary artery disease
Dear reviewers, we would like to thank you first and foremost for your valuable cooperation and supplementary comments on the further development of this work.
We have made all changes to the original document, and all changes have been assimilated with Track Changes.
We will address the individual comments of the two reviewers below.
1st Reviewer Answers
We appreciate all your comments, which are extremely relevant and enriching to our work.
Regarding the comment on ejection fraction and since NYHA, all had preserved ejection fraction and class I. We did not exclude patients with atherosclerosis, and we did not evaluate mean intimal thickness for the entire sample, so these data were not presented.
Regarding the section on medications, this parameter was controlled and no vasodilators were taken. We have revised this information in the article (please see the attachment).
Kind Regards
Authors

Reviewer 2 Report
I am grateful for the opportunity to review this manuscript titled “Arterial stiffness following endurance and resistance exercise sessions in older patients with coronary artery disease”. The purpose of this study was to compare the time course of changes in local and regional indices of arterial stiffness after endurance exercise training and resistance training in the cardiac rehabilitation of patients with coronary artery disease. The data collected in this study may affirm or expand on available literature.
This study is of interest to the IJERPH readers and seems to provide some new findings, applicable to the fields of rehabilitation and training. However, the points mentioned in the “Specific comments” section below should be considered and the manuscript amended accordingly before being considered for publication.
Specific comments
Abstract
1. It would be appropriate for authors to introduce statistical values in the abstract (i.e., p-value, effect size, ...).
Keywords
2. It is recommended that the keywords be different from those included in the title.
Introduction
3. The introduction is well written, but authors must use the introduction to highlight the contribution of their work to the field. In addition, there is little information about the exercise variables that can affect arterial stiffness when both endurance and resistance training are used. Therefore, I encourage the authors to expand this section with these points in mind.
4. The study hypothesis is missing.
Materials and Methods
2.2. Study Design
5. Were the same researchers who performed the assessments? Furthermore, it is not mentioned whether the experimenters were blinded to the research question.
2.3. Exercise Sessions
6. Why did the authors use HR to establish training intensity if patients were taking beta-blockers? It would be more convenient to get the ventilatory thresholds from the treadmill test and prescribe the exercise based on the ventilatory threshold 2. For example, they could use the speed or RPE at which the threshold was reached.
7. How was the load calculated for the resistance exercises? Was a 1RM test performed days before the experiment?
8. It would be interesting if the authors could describe the exercises used in resistance training in more detail.
9. Were both exercise units supervised by specialist staff?
10. Which HR was used for the analysis? the average of the whole session, the maximum, ... Please clarify.
2.4. Central and Peripheral Arterial Stiffness Indices
11. Line 128-129: Please change “Right carotid artery stiffness was measured in the supine position after at least 15-minutes of rest before and within 5 and 15-minutes after each session” to “Right carotid artery stiffness was measured in the supine position after at least 15-minutes of rest before and within 5, 15 and 30-minutes after each session”.
12. Why did the authors only measure for 30 min? The authors could have repeated the measurements at 6, 12, or 24 hours. This would be more clinically relevant.
2.5. Statistical Analysis
13. Authors should describe Æž2.
Results
14. Figure 1. Indicate which point represents EES and RES.
Discussion
15. It is appreciated that the authors have included some limitations of the study. But, in my opinion, they should also include that they limited the measurements to 30 min and did not use a longer period (e.g., 12 or 24 hours after the training session). This data would have more clinical relevance.
Conclusions
16. I recommend that the authors expand on this section. They could add practical applications.
Author Response
Arterial stiffness following endurance and resistance exercise sessions in older patients with coronary artery disease
Dear reviewers, we would like to thank you first and foremost for your valuable cooperation and supplementary comments on the further development of this work.
We have made all changes to the original document, and all changes have been assimilated with Track Changes.
We will address the individual comments of the two reviewers below.
2nd Reviewer Answers
We appreciate all your comments, which are extremely relevant and enriching to our work.
Regarding the comment about introducing statistical values in the abstract, due to the limited word count, we only introduced the p-value for the main interaction effect. However, we have already provided the effect size. Regarding the keywords, I agree and have already changed this.
We appreciate the contribution about the introduction. We tried to explain and present the effects of these two types of training at AR, but there was not much background literature for CAD patients or for acute adaptations. The hypothesis was developed and included in the text.
On methods, the study was always conducted by the same researchers, and regarding the use of methods, it is difficult in practice to guarantee the use of thresholds in all patients, but RPE was always used together with HR.
All patients had exercised for at least 6 months, and the RM were performed one week before each session. All sessions were monitored by exercise physiologists. The HR used for analysis was the maximum per session.
Excellent question about the measurement time points, no doubt today after this work I understand the importance of evaluating them over a longer period of time, and it would be a much richer work, but in future studies we will undoubtedly keep this issue in mind, and it will be of enormous importance for scientific knowledge and clinically relevant. We agree with you that this was a limitation of the study that we included as suggested.
We apologize that the illustration deformed the markers, that has already been corrected.
(please see the attachment)
Kind Regards
Authors

Round 2
Reviewer 2 Report
I am grateful that all comments have been taken into account. The manuscript has improved considerably.